# Experimental Investigation of Vibration Analysis on Implant Stability for a Novel Implant Design

**DOI:** 10.3390/s22041685

**Published:** 2022-02-21

**Authors:** Shouxun Lu, Benjamin Steven Vien, Matthias Russ, Mark Fitzgerald, Wing Kong Chiu

**Affiliations:** 1Department of Mechanical & Aerospace Engineering, Monash University, Melbourne, VIC 3800, Australia; ben.vien@monash.edu (B.S.V.); wing.kong.chiu@monash.edu (W.K.C.); 2The Alfred Hospital, Melbourne, VIC 3004, Australia; m.russ@alfred.org.au (M.R.); m.fitzgerald@alfred.org.au (M.F.); 3National Trauma Research Institute, Melbourne, VIC 3004, Australia

**Keywords:** osseointegration implant, structural health monitoring, vibrational analysis

## Abstract

Osseointegrated prostheses are widely used following transfemoral amputation. However, this technique requires sufficient implant stability before and during the rehabilitation period to mitigate the risk of implant breakage and loosening. Hence, reliable assessment methods for the osseointegration process are essential to ensure initial and long–term implant stability. This paper researches the feasibility of a vibration analysis technique for the osseointegration (OI) process by investigating the change in the dynamic response of the residual femur with a novel implant design during a simulated OI process. The paper also proposes a concept of an energy index (the E–index), which is formulated based on the normalized magnitude. To illustrate the potential of the E–index, this paper reports on changes in the vibrational behaviors of a 133 mm long amputated artificial femur model and implant system, with epoxy adhesives applied at the interface to simulate the OI process. The results show a significant variation in the magnitude of the colormap against curing time. The study also shows that the E–index was sensitive to the interface stiffness change, especially during the early curing process. These findings highlight the feasibility of using the vibration analysis technique and the E–index to quantitatively monitor the osseointegration process for future improvement on the efficiency of human health monitoring and patient rehabilitation.

## 1. Introduction

The traditional treatment for transfemoral amputation is the prosthetic socket system, which consists of an artificial socket providing a secure connection between the residual limb and the prostheses for the transfemoral amputee [1,2]. While enormous advances in the material and design have been utilized to improve the comfort and performance of the socket system, there are several major drawbacks, which include skin irritation and pressure sores as compression occurs between the residual limb and socket transfers via the soft tissue, which compromises the quality of life of the amputee [1,3,4,5]. Furthermore, for extremely short or severely wounded residual femurs, socket devices may not be an appropriate treatment to provide adequate stability and prompt osseointegration (OI) [1,6,7].

The state–of–the–art method that has been widely investigated and employed is the use of osseointegrated prostheses, consisting of a metal implant that is inserted into the skeletal system, providing a direct connection between the artificial limb and the residual leg [8,9] without the interference of the soft tissue [6,8,10]. A common material for the implant is titanium alloy due to its excellent biocompatibility and high resistance to corrosion and repeated stress loading [11,12]. Amputees treated with a transfemoral osseointegrated implant (TFOI) require a two–stage surgical procedure to accomplish a successful implantation. During the first stage, the metal implant is inserted into the medullary cavity and left unload for typically six months [2,13]. Afterwards, the loading on the implant will gradually increase until the TFOI is capable of the full weight load bearing. This process is known as the rehabilitation period, which may take up to 18 months [14]. The direct mechanical connection formed during the OI process provides better joint mobility and improves the control of the prostheses compared to the socket system [9,10,14]. Osseointegrated prostheses also offer further advantages, including better bone conditions, reducing the risk of skin infection, and sensory feedback from the ground during the movement [3,6,15,16].

Even though the OI system has significantly eliminated some limitations of the socket system, there are various challenges that remain for osseointegrated implants; in particular, a long rehabilitation is required to conservatively ensure the implant is securely bonded with residual femur [4,13,17]. The OI system requires sufficient implant stability before and during the rehabilitation period to prevent any potential implant breakage and loosening [1,4,14,15]. In addition, the works in [18,19] reported that periprosthetic infections next to the exposed metal surface could lead to implant loosening.

Hence, reliable assessment methods for the OI process are essential to ensure initial and long–term implant stability. Moreover, early recognition of the implant loosening could also aid in the detection of the infection to prevent severe consequences such as amputations and sometimes death. Currently, various examination methods such as clinical X–ray and magnetic resonance imaging are used to assess the in vivo implant stability [2,20,21]. Nonetheless, these conventional methods are known to be subjective and qualitative since their accuracies are mainly based on the interpretation and judgment of the surgeon rather than using quantitative justifications (i.e., the stiffness of the connection) [1,21,22,23,24]. Furthermore, current amputees are also at potential risk of being exposed to radiation [1,25]. Moreover, the applications of these techniques are limited by the measurement accuracy due to the diffraction effect of the metal intramedullary part [21,22,25]. Therefore, there is an increasing need to quantitatively monitor the degree of OI and prevent implant loosening in long–term applications.

Recently, there is significant interest in the structural health monitoring techniques, which are widely used in monitoring the integrity of structures in the OI system, aiming to increase the reliability and accuracy of assessment of the OI and implant stability. Vibration analysis methods have been actively investigated and used as a non–invasive technique to assess osseointegration–related structures, such as dental implants [17,20,22,26] and total hip arthroplasty implants [2,25,27,28,29,30,31]. Furthermore, the research on vibration analysis methods has extended to the field of transfemoral implants, and the dynamic responses of the OI system are utilized to assess the OI process and implant stability using resonance frequency analysis (RFA) and modal analysis [2,20,21,32,33,34].

In the research on dental implant stability conducted by Huang et al. [35], an implant was installed into predrilled cavities of 3.75 and 5 mm in six rabbits to simulate secure–fit and loose–fit conditions, respectively. The result demonstrated that the femur with the loose–fit implant had a lower initial resonance frequency, compared to the one with a secure–fit condition. In addition, during the OI stage, the resonance frequency significantly increased and peaked under both conditions when the implant was fully bonded with the femur. Similarly, the research on the OPRA implant presented by [2] conducted nine tests during the rehabilitation process of a 40–year–old male patient. The result demonstrated a reduction in resonance frequency after the first weight bearing, which indicated a lack of osseointegration. Then, the resonance frequency gradually increased until the patient was capable of a full bodyweight load.

However, several authors suggested that the RFA should be used in combination with other assessment methods to determine the dental implant stability [36,37,38]. The research on dental implants reported by [39,40] demonstrated a poor correlation between RF and bone–implant contact. These findings may hinder the future application of RFA in monitoring the stability of transfemoral OI implants. Moreover, the research conducted by [20,21] showed that the modal analysis is possible to assess the degree of OI, except with using particular modes over a specific frequency range. Hence, there is still a significant need for universal and robust monitoring techniques for the transfemoral osseointegrated implant.

This paper investigates the feasibility of a vibration analysis technique on the OI process by investigating the change in the dynamic response of the residual femur with a novel implant design during a simulated OI process. The novel osseointegrated implant was developed based on the design concept proposed by Russ, Fitzgerald, and Chiu (US20200188140) [41]. This study also proposes the concept of using normalized energy difference to formulate an energy index (E–index), which could quantify the stages of OI. Moreover, this paper also investigates the effect of implant stiffness on the accuracy of the E–index using different intramedullary stem designs.

## 2. Materials and Methods

### 2.1. Specimens

An amputated 133 mm long Sawbone^®^ composite femur specimen, which represents the most common osteotomy level of 250 mm above the knee, was adopted in this experiment. The femur model was prepared with three specifically designed 3D–printed plastic osseointegrated implants. The first implant model, shown in Figure 1a, consists of three components: an extramedullary (EM) strut, an intramedullary (IM) stem, and a stem. The EM strut aimed to provide initial stability when first inserted into the remnant stem. During the rehabilitation period, the bone tissue slowly grew on the surface of the IM stem, forming a secure connection between the femur and implant. The extended base, which was modified from the prostheses stem, provided a striking point for the experiment. The second implant model, shown in Figure 1b, is the same as the first model but with a hollow IM stem of 0.5 mm in thickness. In addition, amputation normally occurs at different locations. The anatomic cross–section of the remnant stem is not exactly same (refer to Figure 2). Therefore, two oval–shaped implant models, which have the same IM stem and extended strut dimension as the triangular implants, were investigated (see Figure 3).

An epoxy adhesive with a setting time (gel time) of 5 min and a fully cured time of 16 h was used to simulate the process of OI [23,24,42,43,44,45]. The application of epoxy adhesive is not an accurate representation of the OI process; however, the change in the material properties as a result of the curing process mimics the behavior during OI [24]. Due to the OI mainly occurring at the surface of the IM stem, the epoxy was applied to the interface between the femur model and the IM stem. The diameter of the IM stem was slightly reduced by 2 mm, (see Figure 4), allowing the adhesive to have a similar thickness of bone tissue coverage required for the in vivo implant [1]. Triangular and oval implant conditions were carried out four and three times, respectively, to investigate the effect on the E–index due to the variance in the curing time, adhesive quality, and cross–section shape. The results of the tests were notated as ‘Shape–Solid–number’ or ‘Shape–Hollow–number’; for example, Tri–Solid–3 represented test 3 under a triangular solid condition.

### 2.2. Experimental Setup

The proximal end of the residual femur was clamped by a 3D–printed vice, which had jaws fitted with the shape of the femur head, providing relatively rigid fixation. An instrumented impact hammer (B&K Type 8206 Hottinger Bruel & Kjaer UK Ltd., Royston, Hertfordshire, UK) was used to provide a torsional moment input for the femur–implant system. The input force was intentionally not recorded, allowing the force variation during the experiment to aid the assessment of the reliability of the measurement technique proposed [46]. Two unidirectional accelerometers (B&K Type 4507 Hottinger Bruel & Kjaer UK Ltd., Royston, Hertfordshire, UK) were attached to the locations S1 and S2, measured in the y–direction as described in Figure 5. The measuring range of the accelerometers was set up before the measuring process started according to the expected values of the vibration. The voltages from the accelerometers were recorded and analyzed by B&K RT Pro Photon (Hottinger Bruel & Kjaer UK Ltd., Royston, Hertfordshire, UK) with a frequency bandwidth of 14.4 kHz and a frequency resolution of 1.125 Hz. The sampling rate of the B&K RT Pro Photon was 32,768 Hz. The spectra were averaged over 10 samples to achieve a good signal–to–noise ratio. The time of the first recorded experimental data is the reference time (0 s). A total of 1140 s of cure time were recorded. The cross–spectrum of the two sensors was obtained at 30 s intervals for the first 300 s and 60 s intervals for the remaining 840 s.

The application of the two–sensor setup was developed in our earlier work [42,46,47] to identify the bending/torsional modes and to determine the integrity of the signal via a coherence function. However, in the work presented, the ability to separate these vibrational modes is not crucial in determining the E–index.

To aid with the visual identification of the dynamic response during the cure time, MATLAB 2020a (MathWorks, Natick, MA, USA) was used to generate the cross–spectrum and colormap with a normalized magnitude on a logarithmic scale. The cross–spectrum and colormap demonstrated the magnitude change along with the frequency at certain time points and continuous time from 0 to 1140 s, respectively. Furthermore, a quantitative parameter: the energy index (E–index) based on the previous research by Ong et al. [48,49], Vien et al. [24,50], and Wing et al. [23,47], was also adopted in this study to reveal the association between vibrational response and stage of OI. The E–index was defined as the area under the magnitude plot within a certain frequency bandwidth, from $f_{0}$ to$\;f_{i}$, relative to the total area ($f_{0}$ to $f_{1}$), as below:(1)$$E\left(t\right)=E_{fi}\left(t\right)/E_{total}\left(t\right)$$
(2)$$E_{f_{i}}\left(t\right)=\int_{f_{0}}^{f_{i}}M^{2}\left(f,t\right)df$$
(3)$$E_{total}\left(t\right)=\int_{f_{0}}^{f_{1}}M^{2}\left(f,t\right)df$$
where M(f,t) is the normalized magnitude at frequency f and cure time t, $E_{total}\left(t\right)$ is the area covered by the cross–spectrum plot from $f_{0}$ to $f_{1}$ at cure time t, and $E_{f_{i}}$ is the area covered by the cross–spectrum plot from $f_{0}$ to $f_{i}\;$at cure time t.

Additionally, the coherence function, which was used to determine the frequency bound of the E–index, is calculated for the two sensors as defined below:(4)$$Coherece=\frac{{\left|\bar{G_{11}\left(f\right)}G_{22}\left(f\right)\right|}^{2}}{G_{11}\left(f\right)G_{22}\left(f\right)}$$
where $G_{11}\left(f\right)$ and $G_{22}\left(f\right)$ are the autospectra of sensors 1 and 2, respectively, and $\bar{G_{11}\left(f\right)}$ is the complex conjugate of $G_{11}\left(f\right)$.

The determination of the upper ($f_{1}$) and lower ($f_{0}$) bound frequency will be discussed in the next section.

## 3. Results

### 3.1. Determination of Upper Frequency Bound

The coherence of the two accelerometers in the frequency ranged from 0 to 14.4 kHz for Tri–Solid conditions against the cure time, as shown in Figure 6. The result showed that the coherence under 10 kHz was general above 0.8, indicating a good signal–to–noise ratio. A similar phenomenon also occurred in the result of the Tri–Hollow and Oval conditions. Therefore, the upper ($f_{1}$) bound frequency was set to 10 kHz to reduce the error caused by the noise.

### 3.2. Triangular–Shaped Implant

#### 3.2.1. Colormap of Magnitude

Figure 7 shows the normalized magnitude development as the function of cure time for the frequency range of 0 to 10 kHz for Tri–Solid–2. During the early curing of the epoxy, the implant was not fully integrated with the femur; the dampening effect of the soft epoxy suppressed the high–frequency modes leading to a flat cross–spectrum, except for several strong vibrational modes located below 1000 Hz. The comparison of the cross–spectrum over the cure time suggested that several peaks in the high–frequency range appeared as the curing of the epoxy, such as the frequencies 3279 Hz and 4124 Hz, indicated in Figure 7a,b. As the cure time increased, the magnitudes of 3279 Hz and 4124 Hz significantly decreased by 4.064 dB and 5.940 dB, respectively, relative to their baseline (at 0 s) magnitude (Table 1). Similar phenomena were also discovered, as shown in Figure 8 and Figure 9, for the implant with a solid and hollow IM stem, respectively. From 0 s to 300 s, the resonance peaks were hard to distinguish. After the first 300 s, the resonance peaks in the frequency band from approximately 2000 to 8000 Hz became noticeable for both solid and hollow implants, especially for the resonance peak within 3000 to 4000 Hz, marked by the dotted line. This was consistent with the five–minute setting time of the epoxy used in this experiment, meaning that the sensitivity of early curing is significant after the initial bonding provided by the adhesive at the interface. The plots also demonstrate that the magnitude of the several modes below 1000 Hz was not affected by the interface stiffness change due to the simulated OI process. Therefore, the lower frequency ($f_{0})$ for the E–index formula was set to 1000 Hz to increase the sensitivity of E–index on the OI–related stiffness change.

#### 3.2.2. E–Index against Frequency

Figure 10 demonstrates the variation in the E–index at five different times (0, 150, 300, 600, and 1140 s) of Tri–Solid–2 for the frequency ranges of 0–14.4 kHz, 0–2 kHz, and 1–10 kHz, respectively. The E–index in the frequency range 1–10 kHz showed a significant change along the cure time; nevertheless, no change in E–index for the other two frequency ranges was demonstrated in the plot. This result indicated that the energy of the vibrational modes within the frequency range 1–10 kHz was sensitive to the stiffness change in the femur–implant interface caused by the simulated OI process.

The plots of the E–index as the function of frequency at five different times (0, 150, 300, 600, and 1140 s) are shown in Figure 11 and Figure 12 for the solid IM stem and hollow IM stem conditions, respectively. For both conditions, the E–index for a frequency range of 2000 Hz to 6000 Hz gradually increased over time and became stable around 1, except for the Tri–Hollow–3 condition, which was stable at 0.41. After 300 s, the gradient of the E–index approached zero as the femur fully bonded with the implant. The difference between 0 s and 1140 s gradually decreased as the frequency increased from 2000 Hz to 6000 Hz. Moreover, based on Figure 8 and Figure 9, which demonstrate significant changes in the magnitude at approximately 3500 Hz, this means that the vibrational modes at this frequency would be more sensitive to the femur–implant interface stiffness change as compared to other frequencies. There was a significant change (at least 57%) in the E–index at the end of the experiments, related to the baseline at 3500 Hz for both solid and hollow conditions, which are recorded in Table 2. Moreover, the averaged difference of the hollow condition was larger when compared to that of the solid condition. Except for this, there was no significant pattern in the E–index difference and frequencies between the solid and hollow conditions. These two conditions shared a similar trend of the E–index over the cure time.

#### 3.2.3. E–Index against Cure Time

The following graphs in Figure 13a,b for the solid and hollow conditions, respectively, were generated based on the E–index formula at 3500 Hz. Figure 13a evidences that for the solid condition, the E–index increased dramatically for the first 300 s, then the gradient of the plots approached zero, and the value stabilized above 0.8 as the implant was initially bonded with the femur. The distinct gradient change around 300 s coincided with the setting time of the epoxy adhesive. The E–index for the hollow condition shared similar behavior to that of the solid conditions. However, for Tri–Hollow–3 in Figure 13b, the E–index showed some fluctuations during and after the adhesive setting time. This result is noticeably different from the other data presented in this paper, which will be discussed in the discussion section. Except for this, the plots of the E–index under the solid and hollow conditions showed a clear trend that could be used to identify the stage of the OI process. The E–index offered a quantitative approach to monitor the OI process rather than one based on visual inspection of the colormap or selecting and identifying the specific frequency peaks/modes on the cross–spectrum.

### 3.3. Oval–Shaped Implant

#### 3.3.1. Colormap of Magnitude

The colormap (refer to Figure 14) for the oval–shaped implant presented a similar trend to that of the result of the triangular implant. A significant magnitude change was identified at 300 s for both the solid and hollow conditions. In addition, the change in the magnitude against cure time for the modes located below 1000 Hz was hard to distinguish. These results coincide with those for the triangular implant, proving that the colormap of the magnitude was not affected by the cross–section of the residual femur. This result evidences the potential of the colormap in detecting the initial bonding of the OI process for different geometry at a fixed length of the residual femur of 133 mm.

#### 3.3.2. E–Index against Cure Time

Similar to the triangular implant, Figure 15 presents the E–index comparison between each specimen over the cure time. It is shown that the E–index steadily increased along the cure time, being stable around 0.9 at 300 s as the implant bonded with the femur. The curing of the adhesive resulted in a relatively high gradient for the first 300 s. There was an increase (at least 28%) in the E–index over the curing time for both the solid and hollow conditions, which are recorded in Table 3. However, the gradient of the E–index of the solid condition was smaller compared to that of hollow condition. In addition, the difference in the E–index relative to 0 s indicated that the hollow condition had a larger change in the E–index compared to that of the solid condition, which was coincident with the result of the triangular condition. This can be attributed to the fact that the similar stiffness change caused by the adhesive curing was relatively smaller in the solid implant system due to the additional stiffness of the solid IM stem. These reductions in the stiffness change between solid and hollow implant system implied that the E–index was sensitive to the stiffness of the femur–implant system, evidencing that the E–index could be a quantitative parameter to determine the degree of OI.

## 4. Discussion

The variation in the magnitude on the colormap revealed that during the early curing of the epoxy, the implant was not fully integrated with the femur; the vibration of the system was hindered by the high dampening effect of the adhesive, leading to a flat cross–spectrum, except for several strong fundamental vibrational modes located below 1000 Hz, which were not affected by the OI. Due to the increase in stiffness at the femur–implant interface provided by the adhesive, several peaks above 2000 Hz appeared as the curing of the epoxy, leading to an increase in the E–index, shown in Figure 11 and Figure 12. These results indicate that with the additional axis of the time variation, the colormap could be used to visually inspect the change in the magnitude along with the simulated OI process. More importantly, the result from the oval implant indicated the colormap possesses favorable universality for different cross–section geometries of the residual femur. Based on the appearance of the resonance peaks and the clear step change in the magnitude, the colormap is capable of identifying the change in the femur–implant interface condition, thereby indicating the initial stage of OI.

The result of the E–index over time indicated that E–index was sensitive to the interface stiffness change, especially during the early curing process. This finding suggested that the E–index could potentially be used as a quantitative justification to help with the detection of the initial stage of the OI process. Moreover, the E–index comparison between the solid and hollow conditions indicated the difference in the E–index due to the additional stiffness provided by the solid IM stem, illustrating that the E–index was sensitive to the stiffness change. In addition, the result of the oval–shaped implant demonstrated the capability of the E–index in monitoring the OI process without selecting or identifying the most sensitive resonance frequency based on the geometry of the residual femur. However, there were some fluctuations in the E–index over the cure time. The potential cause of this might be due to the adhesive epoxy not being properly mixed or the adhesive layer not being applied uniformly, which consequentially results in uneven curing at different areas of the IM stem. This added variations in the full cure time and adhesive quality. Furthermore, the maximum bond time for this glue is around 16 h, which is much larger than the experiment time of 19 min. This means that the manual excitation may introduce damage to the interface before the connection reaches its full strength, leading to an unstable E–index.

After the experiment, the implant model was removed from the femur to inspect the condition of the adhesive layer between the femur and implant. Upon inspection, there were clear differences in glue fracture markings between Hollow–3 and the other specimens after implant removal. The surfaces of those sufficiently bonded had indications of bonding contacts with glue shear marks, whereas, in Hollow–3, there were minimal indications of these marks (refer to Figure 16). Furthermore, there is evidence that the glue seeped into the medullary cavity and, as a result, did not provide adequate bonding due to an insufficient amount of glue. Nevertheless, this represents a real–life scenario of insufficient OI. According to Figure 13, the E–index with sufficient bonding presented a distinct gradient change around 300 s and was stable above 0.8. However, Tri–Hollow–3 demonstrated irregular fluctuations and was unable to reach a steady value of the E–index along the stimulated OI process. Even though the result of Tri–Hollow–3 showed a clear increase in the level of E–index related to the baseline after 300 s, the inadequate bonding caused by less adhesive, as compared to the other conditions, was identified in the E–index with a clear reduction in value from above 0.8 (which was anticipated as sufficient bonding) to 0.41. In addition, this insufficient interface stiffness was not capable of providing sufficient resistance to the manual excitation, resulting in damage in the connection, which represented a large variation in the E–index, as shown in Figure 13b. These significant differences in the E–index evidenced that the E–index is able to identify insufficient implant stability and that it is also capable of dynamically monitoring the femur–implant connection during the OI process. Nevertheless, future work will validate the robustness of this strategy by simulating common implant and interface failures.

Overall, the results displayed a clear trend of the E–index along the stimulated OI process, evidencing the potential of the E–index as a quantitative approach for monitoring the stages of OI. Additionally, the similar trend between the triangular and oval conditions suggests that the E–index is less affected by the geometry of the residual femur’s cross–section. Compared to the RFA [2] and modal analysis [20] for the femur–implant system, the E–index quantified the change in the system’s energy for a fixed frequency range during the OI process rather than selecting and identifying the specific frequency peaks/modes for a certain implant design or length of the residual femur. In addition, the E–index demonstrated a large increase of 80% and 50% for the triangular– and oval–shaped implants, respectively, compared to the 3% for the RFA [2] and 10–47% difference and modal analysis [20]. The application of the E–index could be possibly employed as a universal way to assess the OI process for various implant designs and remaining lengths. However, further investigation of the E–index on various lengths of the residual femur and clinical studies are required to develop a robust monitoring technique for the femur–implant system.

The work presented here has outlined the potential of determining implant stability with a sensor located on the external portion of the implant. However, we expect our future work to include an investigation into potential sensing techniques by embedding sensors on the extramedullary struts and/or intramedullary stem, which could assess implant stability under in vivo conditions.

## 5. Conclusions

A vibration analysis method has been demonstrated in this paper to assess novel implant stability and monitor the integration of the femur and the implant. The colormap of normalized magnitude against time revealed changes in the resonance peaks at the frequency band of 2000 Hz to 8000 Hz as the adhesive cured, identifying the beginning of the stimulated OI process. Moreover, the energy method mentioned in this paper, the E–index, increased by 80% and 50% for the triangular– and oval–shaped implants, respectively, along the cure time related to the time of 0 s during the experiment. These results indicate that the introduction of the E–index quantified the bonding quality between the femur and the implant for continuous monitoring of the OI process. This demonstrates the feasibility of this novel vibrational method for monitoring the OI process. Further work is currently in progress to further investigate the accuracy and reliability of the E–index.

## Figures and Tables

**Figure 1 sensors-22-01685-f001:**
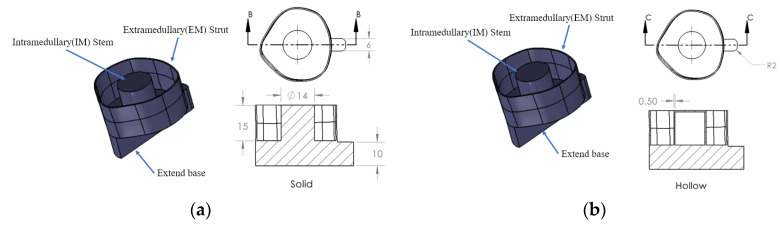
Triangular–shaped implant model and dimension developed based on Patent US20200188140: (**a**) implant with solid IM stem and (**b**) implant with hollow IM stem.

**Figure 2 sensors-22-01685-f002:**
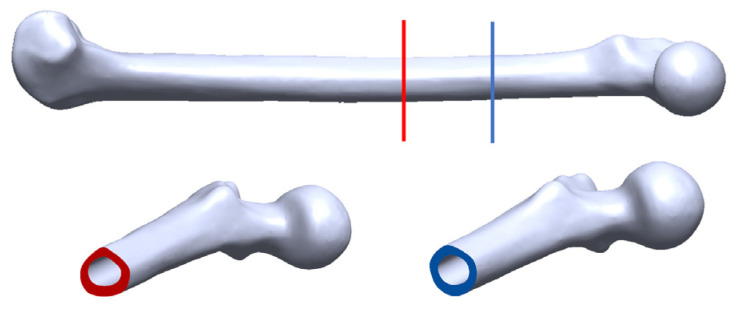
3D femur model demonstrating the triangular and oval cross–section for different anatomic locations.

**Figure 3 sensors-22-01685-f003:**
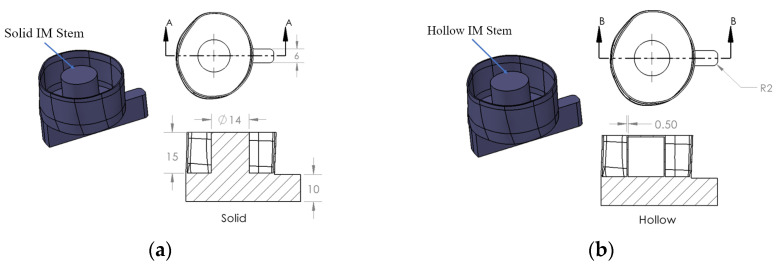
Oval–shaped implant model and dimension developed based on Patent US20200188140: (**a**) implant with solid IM stem and (**b**) implant with hollow IM stem.

**Figure 4 sensors-22-01685-f004:**
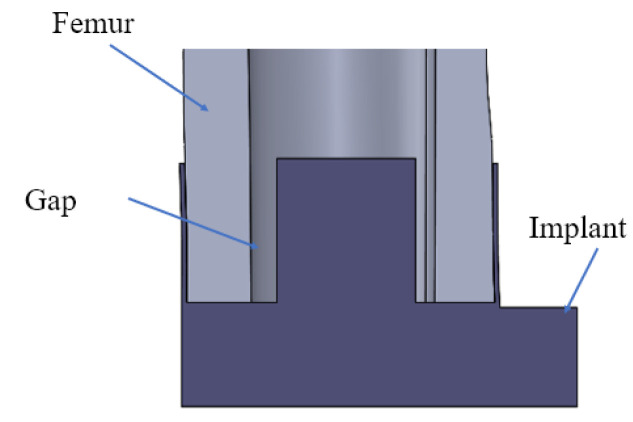
Cross–section of femur–implant interface.

**Figure 5 sensors-22-01685-f005:**
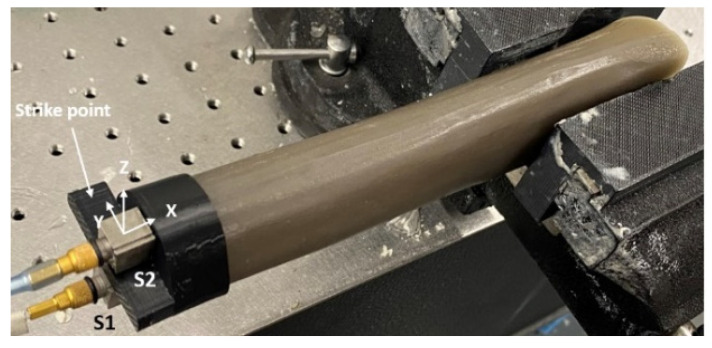
Image of experimental setup with the orientation of sensors.

**Figure 6 sensors-22-01685-f006:**
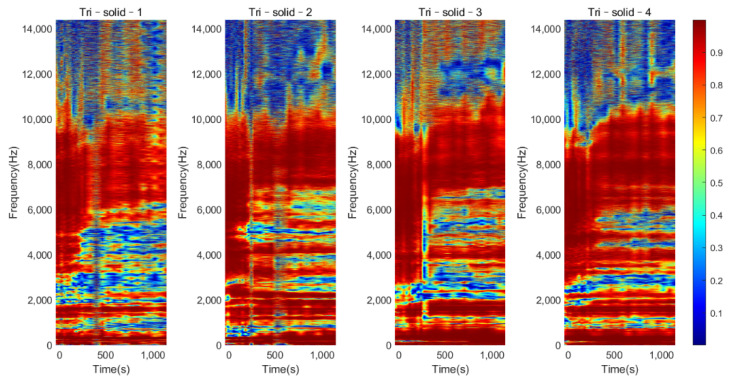
Plots of the coherence development against the cure time for triangular implants with a solid IM stem.

**Figure 7 sensors-22-01685-f007:**
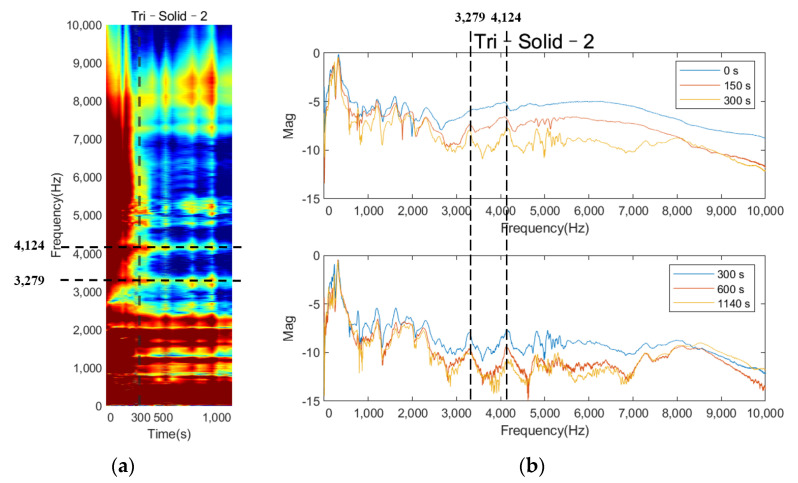
Plots of (**a**) colormap based on normalized magnitude development as the function of cure time, (**b**) cross–spectrum at five different cure times.

**Figure 8 sensors-22-01685-f008:**
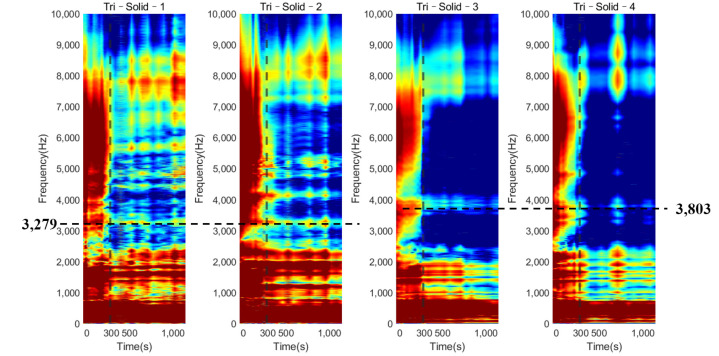
Plots of the normalized magnitude development as the function of cure time for triangular implants with a solid IM stem.

**Figure 9 sensors-22-01685-f009:**
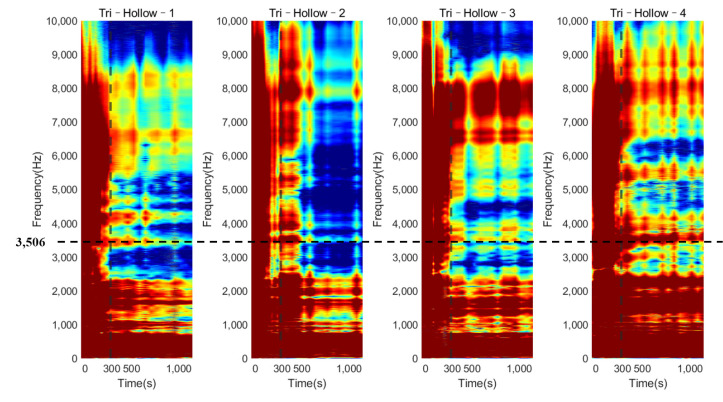
Plots of the normalized magnitude development as the function of cure time for triangular implants with a hollow IM stem.

**Figure 10 sensors-22-01685-f010:**
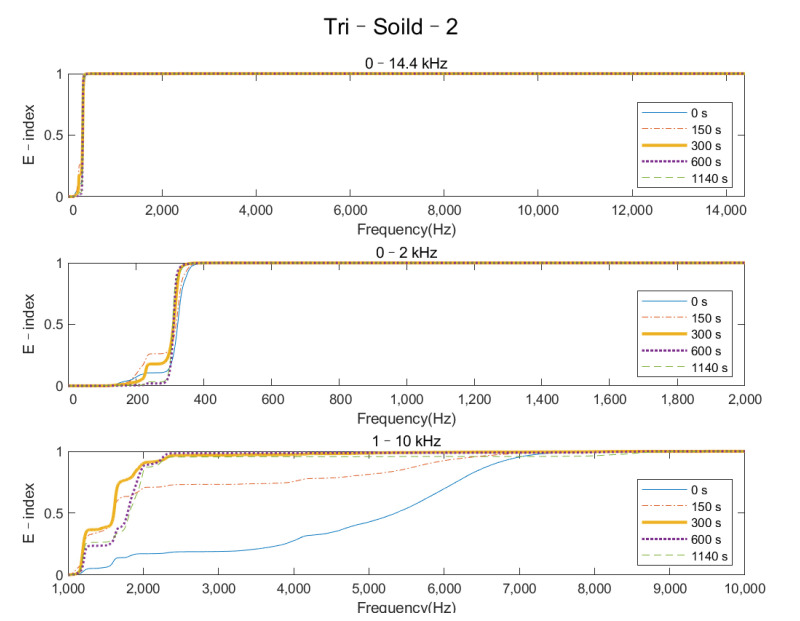
Plots of the E–index of triangular implants with a solid IM stem at five different times (0, 150, 300, 600, and 1140 s) for frequency ranges of 0–14.4 kHz, 0–2 kHz, and 1–10 kHz.

**Figure 11 sensors-22-01685-f011:**
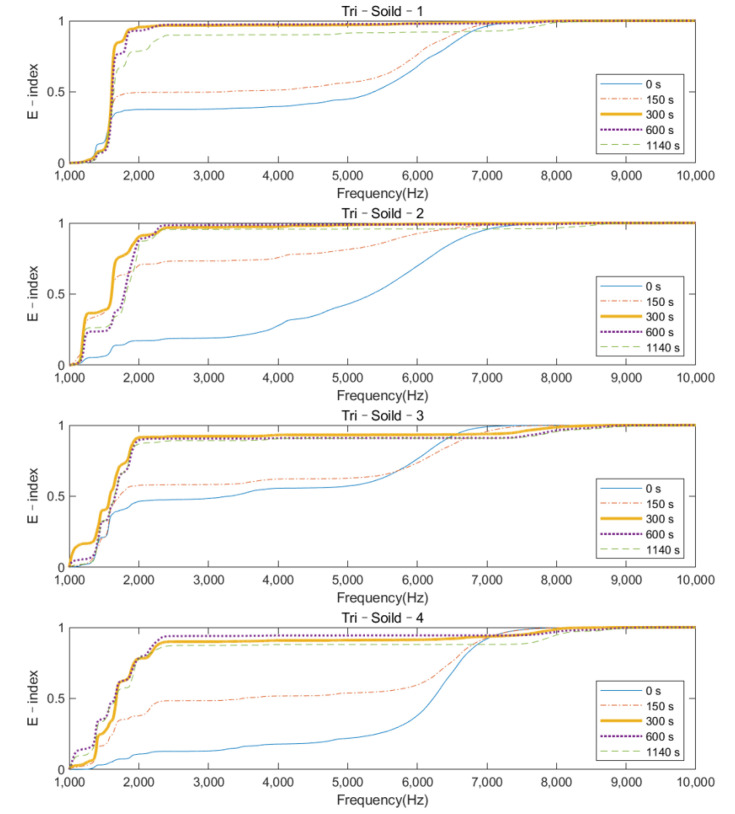
E–index as the function of frequency at five different times (0, 150, 300, 600, and 1140 s) for a solid IM stem condition.

**Figure 12 sensors-22-01685-f012:**
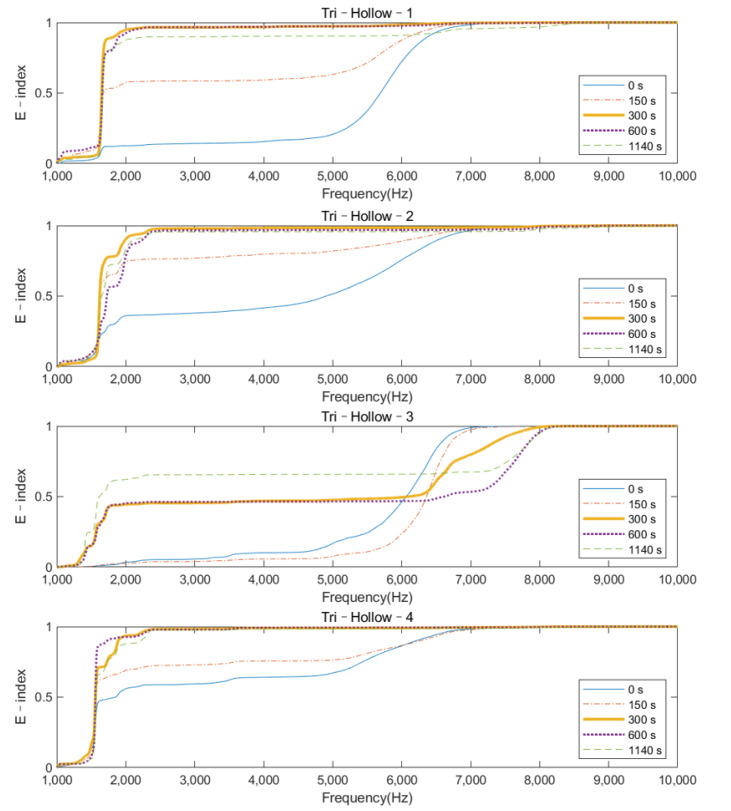
E–index as the function of frequency at five different times (0, 150, 300, 600, and 1140 s) for a hollow IM stem condition.

**Figure 13 sensors-22-01685-f013:**
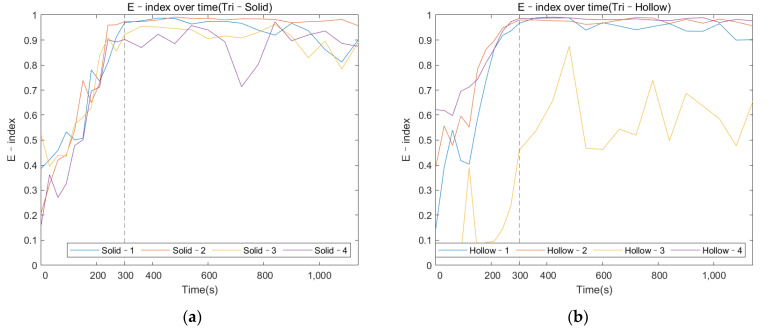
Variation of E–index at selected frequencies as the cure time increased for the triangular implants under (**a**) solid and (**b**) hollow conditions.

**Figure 14 sensors-22-01685-f014:**
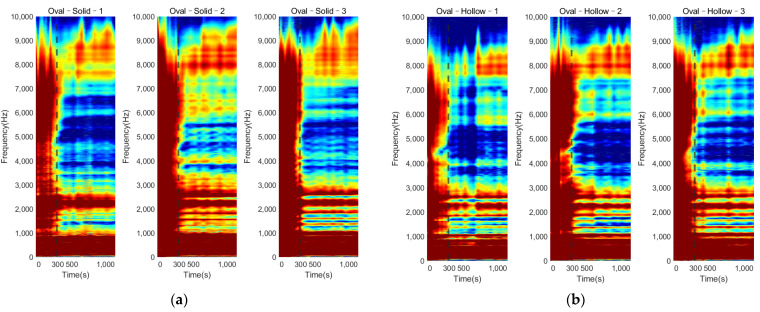
Plots of the normalized magnitude development as the function of cure time for oval implants with (**a**) solid IM stem and (**b**) hollow IM stem.

**Figure 15 sensors-22-01685-f015:**
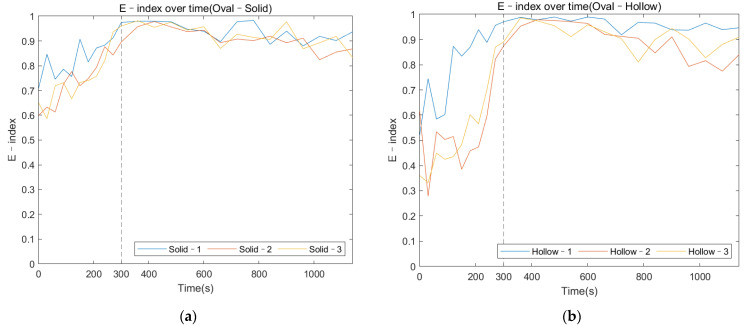
Variation of E–index at selected frequencies as the cure time increased for the oval implants under (**a**) solid and (**b**) hollow conditions.

**Figure 16 sensors-22-01685-f016:**
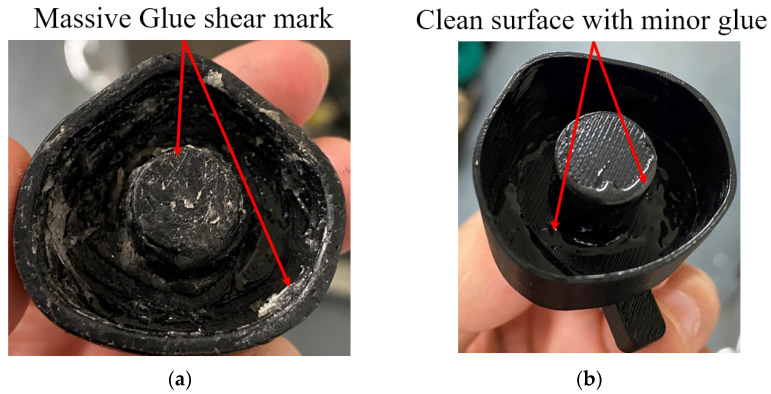
Implant surface with (**a**) adequate bonding and (**b**) insufficient bonding.

**Table 1 sensors-22-01685-t001:** Normalized magnitude in log scale for frequencies of 3279 and 4124 Hz.

Frequency (Hz)	0 s	150 s	300 s	600 s	1140 s	Change in dB (Relative to 0 s)
3279	−6.010	−7.425	−7.965	−9.623	−10.074	−4.064
4124	−5.195	−6.684	−7.610	−9.391	−11.135	−5.940

**Table 2 sensors-22-01685-t002:** Difference (relative to 0 s for each condition) at 3500 Hz.

Test	Difference (%)	
	Solid	Hollow
1	132.82	516.53
2	354.55	142.80
3	71.81	713.73
4	439.45	57.17
Averaged	249.66	355.04

**Table 3 sensors-22-01685-t003:** Difference (relative to 0 s for each condition) for oval–shaped implants.

**Test**	**Difference (%)**	
	**Solid**	**Hollow**
1	32.05	81.53
2	44.93	36.46
3	28.11	152.37
Averaged	35.03	90.12

## Data Availability

The raw/processed data required to reproduce these findings cannot be shared at this time as the data also forms part of an ongoing study.

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
