# Peer review of "Experimental Investigation of Vibration Analysis on Implant Stability for a Novel Implant Design"

_sensors, 2022, doi:10.3390/s22041685_

Round 1
Reviewer 1 Report
Please check the attached pdf file.

Reviewer 2 Report
This article is interesting; however, the authors need to take in consideration the following suggestions before accepting it:
- The introduction needs to be improved by discussing current articles because only a few new articles have been included in it.
 - The quality of figures needs to be increased, also the size of labels needs to be increased.
 - It is important to justify the selected feature (energy) in your work, why do not other features such as variance, RMS, etc have been tested?.
 - The authors need to explain better the experimental setup, or how were obtained the signals, i.e., sampling frequency, etc.
 - The authors do not compare their results with other methods proposed in the literature recently in order to validate or show that their contribution is important in the subject.
 - The conclusion needs to be improved, adding quantitative results not only qualitative results. In addition, it is important to mention, what is the next with the investigation?

Reviewer 3 Report
This paper presents an xperimental study on the vibration analysis of a novel implant design. This is a interesting paper but it is lack of the theoretical background. I strongly recommend the authors to include this part.
Round 2
Reviewer 1 Report
My questions have been well answered in the authors' responses. Also, the manuscript quality is improved with a more detailed background introduction and references, with the help of the other reviewers. Therefore, I decided to recommend the manuscript publishing on Sensors.
Author Response
We wish to thank the reviewers for their comments on our manuscript “Experimental Investigation of Vibration Analysis On Implant Stability For a Novel Implant Design”. Thank you for the thorough and critical review, the thoughtful comments, and the constructive suggestions, which help to improve this manuscript.
Reviewer 2 Report
The authors must justify analytically or experimentally the use of energy as a feature in your proposal and it is superior to other features. In addition, it is mandatory that the authors provide a comparison with other proposals in order to observe the advantages and disadvantages of their proposal.
